# Fungal Endophytes and Their Role in Agricultural Plant Protection against Pests and Pathogens

**DOI:** 10.3390/plants11030384

**Published:** 2022-01-30

**Authors:** Rachel Grabka, Tyler W. d’Entremont, Sarah J. Adams, Allison K. Walker, Joey B. Tanney, Pervaiz A. Abbasi, Shawkat Ali

**Affiliations:** 1Kentville Research and Development Centre, Agriculture and Agri-Food Canada, Kentville, NS B4N 1J5, Canada; 142077g@ACADIAU.CA (R.G.); pervaiz.abbasi@agr.gc.ca (P.A.A.); 2Department of Biology, Acadia University, Wolfville, NS B4P 2R6, Canada; tylerdentremont@acadiau.ca (T.W.d.); sarah.adams@acadiau.ca (S.J.A.); allison.walker@acadiau.ca (A.K.W.); 3Pacific Forestry Centre, Canadian Forest Service, Natural Resources Canada, 506 Burnside Road West, Victoria, BC V8Z 1M5, Canada; joey.tanney@NRCan-RNCan.gc.ca

**Keywords:** endophytic fungus, plant protection, biocontrol, antagonism, defence activation

## Abstract

Virtually all examined plant species harbour fungal endophytes which asymptomatically infect or colonize living plant tissues, including leaves, branches, stems and roots. Endophyte-host interactions are complex and span the mutualist–pathogen continuum. Notably, mutualist endophytes can confer increased fitness to their host plants compared with uncolonized plants, which has attracted interest in their potential application in integrated plant health management strategies. In this review, we report on the many benefits that fungal endophytes provide to agricultural plants against common non-insect pests such as fungi, bacteria, nematodes, viruses, and mites. We report endophytic modes of action against the aforementioned pests and describe why this broad group of fungi is vitally important to current and future agricultural practices. We also list an extensive number of plant-friendly endophytes and detail where they are most commonly found or applied in different studies. This review acts as a general resource for understanding endophytes as they relate to potential large-scale agricultural applications.

## 1. Introduction

In agriculture, plant pathogens and pests reduce the global annual crop yield by an estimated 30 to 50%, which is a loss that must be combatted to ensure food security for an ever-increasing human population [1]. These organisms are usually controlled by chemical pesticides to reduce the crop loss and fulfill the food demand. However, recent restrictions on different chemical pesticides and an increased consumer demand to reduce chemical pesticide residues in both the food supply and environment are urging both governments and private agriculture industries to pursue alternative, clean technologies for plant production [2,3,4,5]. One underexplored, but promising, alternative approach is gaining attention: the use of beneficial endophytes as biological control agents for crop protection [5,6,7,8,9,10]. Endophytes are microorganisms that live inside the plant for all or part of their life cycle while not causing damage or disease symptoms in their host most of the time [10,11]. Almost all vascular plants examined to date harbor endophytes that are believed to originate in the rhizosphere and phyllosphere and enter the host plant through natural openings or wounds. In recent years, many studies have explored the endophytic communities associated with different plant species [12,13,14,15,16,17,18,19,20,21,22,23]. These studies have shown that the diversity of fungal endophytes that reside inside plants is largely underestimated. It has also been shown that the distribution of some endophytes is host and/or environment specific [20,24].

Endophytic microorganisms promote plant growth and provide protection against pests and pathogens through different mechanisms [18,19,25]. Endophytes produce and secrete secondary metabolites/biochemicals that suppress/reduce the negative effects from plant pathogens, including volatile compounds that are able to suppress pathogen growth [26]. Other endophytes protect their host plant by inducing plant defence mechanisms [27], which can be achieved by systemic acquired resistance (SAR) or induced systemic resistance (ISR) [19,28]. An example of a host-induced defence mechanism is *Piriformospora indica*, inducing a jasmonic acid-dependent defence response in *Arabidopsis thaliana* by co-inoculation with a pathogen [29]. Some endophytes may demonstrate their biocontrol potential by secreting antifungal and antibacterial compounds, thereby inhibiting the competition of pathogens, or they may exhibit mycoparasitic activity (i.e., parasitism of one fungus by another) [9]. Recently, it has been shown that an *Enterobacter* sp. strain isolated from finger millet (*Eleusine coracana*) is able to suppress the grass pathogen *Fusarium graminearum* in the root system of its host plants and simultaneously produces several antifungal compounds that kills the fungus [30]. Endophytes also directly compete with the host pathogens for space and nutrients [31,32]. Foliar application of endophyte-free leaves of *Theobroma*
*cacao* with a mixture of endophytes protected against leaf necrosis and leaf mortality in leaves challenged with a *Phytophthora* sp. [33]. This protection was localized in inoculated leaves and could not be readily correlated with in vitro endophyte interactions, suggesting that complex interspecific interactions (such as competition and mutual antagonism) may play an important role in mediating host defence outcomes.

In addition to protecting their host plants against pathogens directly, several endophytes have plant growth promoting (PGP) properties that result in a stronger plant. These PGP endophytes not only provide nutrients such as nitrogen, phosphate and/or iron, but can facilitate plant growth and development by growth stimulation [34]. Associated with roots, PGP microbes can produce several chemical compounds that influence plant growth and development. These include the plant hormones indole-3-acetic acid (IAA), gibberellins, and cytokinins, and/or 1-aminocyclopropane-1-carboxylic acid (ACC) deaminase activity [35,36]. The latter was shown to promote plant mycorrhization [37]. Endophytes can also modulate plant hormones such as auxin, cytokinin, ethylene and gibberellin, and produce other bioactive compounds [38,39]. These PGP microbes can play an indirect role in plant protection against pathogens and pests by improving growth and overall health of their hosts compared to non-colonized counterparts.

Fungal endophytes are asymptomatic inhabitants of plant tissue and are reported from all parts of plants [40,41,42]. A plant may harbour numerous endophytic species, which may remain localized and lead to tissue-specific protection from disease [42,43] or can spread systemically in herbaceous plants [44,45]. These symbiotic, and potentially mutualistic, interactions between plants and endophytes are diverse and span both wild and cultivated plant species [46]. In almost every instance, examining host plants reveals the presence of endophytes [1]. The ubiquitous nature of endophytes is increasingly a focus in plant-fungal studies, which have traditionally focused on phytopathogenic or mycorrhizal fungi [46]. More than 1 million endophytic species are estimated to exist in 300,000 different plant species, but only a small fraction have been isolated and investigated for their roles within the plants they inhabit [47].

Of those that have been studied, some endophytes can offer a range of benefits to their plant hosts, offering an increase in plant fitness over uninhabited counterparts [48,49]. Endophytes can alleviate abiotic and biotic stressors such as drought, salinity, heavy metals and other toxic compounds introduced by the environment, flood, extreme temperatures, predators and pathogens [49,50]. Endophytes provide beneficial biological properties to the hosts, such as deterring pathogenic microbes, insects and other herbivores, while also providing stimulants for plant growth and development [51]. As plant pathogens and pests are well known for reducing global crop yield by an estimated 30 to 50% annually [1], endophytes, whose beneficial properties can improve plant fitness and crop yield while still maintaining quality and safety, represent a notable avenue in combatting plant loss.

In this review, we focus on the important roles fungal endophytes play in protecting agricultural crops against common pathogens and non-insect pests such as fungi, bacteria, nematodes, viruses, and mites. We also report on the effects of the environment and host plant feedback on fungal endophytes and explore endophyte transmission between hosts (horizontal) as well as inherited (vertical) transmission.

## 2. Fungal Endophytes and Their Effects on Fungal Pathogens

Fungal pathogens cause some of the most devastating damage to crops by killing plants, reducing yield and quality, and causing postharvest losses [1]. Some fungal pathogens also produce mycotoxins that are detrimental to the health of humans and livestock [1]. Synthetic chemical fungicides have become a mainstay in agriculture to control fungal pathogens; however, like other pesticides, fungicides can have detrimental non-target impacts on the environment, for example on fungi beneficial to crop health [52,53,54,55,56]. For instance, extensive fungicide use impacts mutualist fungi such as arbuscular mycorrhizae, whose loss can lead to dramatic decreases in plant fitness [52]. Fungicides can also selectively harm non-target beneficial microorganisms over pests [57]. Biocontrol endophytes, such as *Ampelomyces*, one of the first biocontrol fungi used against pathogenic fungi, are environmentally friendly alternatives to chemical fungicides, decreasing pathogen prevalence while maintaining mutualistic fungi. As biocontrol endophytes are capable of reducing adverse environmental effects of chemical fungicides [58], the inclusion of such biocontrol agents in integrated pest management approaches can improve sustainability in the agricultural sector and maintain or even enhance soil health. In addition, applying diverse pest management strategies may also reduce the occurrence of, or manage for, chemical pesticide resistance.

Secondary metabolites produced by endophytes are being extensively studied with the goal of identifying natural products that are useful as agrochemicals [59,60]. Top-down approaches have been used to extract and isolate diverse compounds from selected taxonomic orders of fungi [59]. A recent review of compounds produced by *Xylariales* highlights the exceptional diversity of bioactive metabolites that have been isolated from species within this order, including glucosides, cytochalasans, azaphilones, terpenoids, non-ribosomal peptides, macrolide polyketides, benzenoids and lactones [59]. Other studies use a different approach to determine the specific antifungal compounds that may control plant pathogens. In these studies, the fungal endophytic diversity of a host plant species is characterized, and endophyte cultures are selected for dual culture assays to assess antagonism against known pathogens of the host plant [44,61,62,63,64]. Antagonism against pathogens is primarily determined by the presence of inhibition zones between the endophytic and pathogenic fungi, or the ability of the endophytic fungi to overgrow the pathogenic fungi [61,62,65,66,67]. Cultures showing anti-pathogen activities undergo compound extraction and analysis with liquid chromatography or gas spectrometry run in tandem with a mass spectrometry [59,60,68]. The results of such studies aid in the identification of candidate endophyte species, which can be further investigated for their biocontrol potential, and consistently show that antagonists of pathogens are an inherent part of the plant microbiome [61,62,69,70]. The antifungal activities of compounds produced by some endophytes have been studied for their mode of effectiveness against several different pathogenic fungi and their ability to increase host plant fitness [44]. In many cases, however, the mechanism of how these endophytes provide such benefits to their host remains elusive or understudied [44].

Endophytes can also enhance host plant resistance to fungal pathogens by inducing a systemic response after endophytic colonization [71,72]. The plant initiates a defensive strategy using cell wall deposits to strengthen cell walls and defend them from penetration [71]. Endophytes possess mechanisms such as exoenzymes to allow them access to these strengthened cells, but the deposits may prevent pathogens from doing the same [71]. Endophytes can also act as priming stimuli that induce plant defence responses through transcriptional reprogramming; for example, by modulating the expression of downstream defence-related genes such as those involved in salicylic acid, jasmonic acid and ethylene signaling pathways [70,73,74,75,76]. Colonization by endophytes (and pathogens) and subsequent metabolite secretion have also been associated with increasing the rate of photosynthesis (*Sclerotinia sclerotiorum*), chlorophyll content of plant cells, density of trichomes and stomata on plant tissues (*Beauveria bassiana*), antioxidant enzyme activity, callose deposition, cell lignification and phytoalexin accumulation (*Diaporthe liquidambaris*) [70,73,77]. Along with these modes of protection, competitive exclusion between endophytes and pathogenic fungi may occur [32,72]. Competitive exclusion describes the general suppression of pathogen establishment by endophytes colonizing and occupying the same potential niche. This method of protection can occur in the absence of the aforementioned mechanisms.

Fungal endophytes from the genus *Daldinia* inhibit the growth of the plant pathogens *Colletotrichum acutatum* and *Sclerotium rolfsii* [78,79]. *Daldinia eschscholtzii* isolated from ginger, *Zingiber officinale*, and Stemona root, *Stemona tuberosa*, was found to produce 60 identifiable compounds, the major ones being elemicin (24%), benzaldehyde dimethyl acetal (8%), ethyl sorbate (7%), methyl geranate (6%), trans-sabinene hydrate (5%) and 3,5-dimethyl-4-heptanone (5%) [79]. Elemicin is reported as an effective antifungal against *Colletotrichum gloeosporoides*, *C. nymphaeae* and *C. musae* [79]. *Daldinia* cf. *concentrica* isolated from olive, *Olea europaea*, produced 27 volatile organic compounds (VOCs), including 3-methyl-1-butanol, (±)-2-methyl-1-butanol, 4-heptanone, isoamyl acetate and trans-2-octenal [80]. Solutions containing mixtures of these VOCs showed a broad spectrum of antifungal activities [80]. *Daldinia* spp. have also been reported to produce the antimicrobial compounds α-guaiene, guaia-1(10), 11-diene, (−)-à-Panasinsen and thujopsene [81].

The genus *Fusarium* contains many species known as both plant pathogens and endophytes capable of inhibiting other fungal pathogens [82]. Many studies have investigated *Fusarium* metabolites for their application as pharmaceutical antimicrobial agents, but less focus has been placed on the antifungal properties of these compounds and their application in agricultural systems [82]. A crude extract of *F*. *proliferatum,* isolated from the medicinal plant *Cissus quadrangularis*, inhibited the growth of *Rhizoctonia solani* and *F*. *oxysporum* at concentrations of 0.2–2.5 mg/mL [83]. Further analysis of the crude extract revealed that it contained phenolics, terpenoids and unsaturated alkenes [83]. *Fusarium chlamydosporum* chitinase, once purified, was found to lyse cell walls of germ tubes and urediniospores of the rust species *Puccinia arachidis* and subsequently prevented urediniospore germination [84].

Other endophytes investigated for the antifungal properties of their secondary metabolites include species from the genera *Aspergillus*, *Colletotrichum, Diaporthe, Gliocladium, Lecanicillium*, *Phyllosticta* and *Trichoderma*. *Trichoderma asperellum*, *T*. *atroviride* and *T. longibrachiatum* isolated from soybean (*Glycine max*) were shown to reduce infection of seeds by the pathogen *Rhizoctonia solani* by 64, 60 and 55%, respectively, when applied in solution to infected soils [85]. The *Trichoderma* species produced the hydrolytic enzymes pectinase and chitinase, all capable of degrading cell wall components [85]. Additionally, the *Trichoderma* species produced siderophores, which reduce the availability of iron to pathogenic fungi, and IAA, which has a strong effect on plant growth [85]. *Trichoderma erinaceum* isolated from ginger and Stemona root was shown to inhibit the growth of the southern stem rot disease agent *Sclerotium rolfsii* by 64% in dual culture assays and reduce infection by 58% in pot experiments [78]. Extract analysis determined that *T*. *erinaceum* produced the polyketide group 6-n-pentyl-2H-pyran-2-one (6PAP), β-1,3 glucanase and chitinase, which had inhibitory effects on the growth of *S*. *rolfsii* [78].

Extracts from *Aspergillus neoniger* isolated from the medicinal plant *Ficus carica* were found to inhibit the growth of pathogens *Penicillium avelaneum*, *P. notatum* and *A. terreus* by 80% or more [86]. Analysis by high-performance liquid chromatography and nuclear magnetic resonance spectroscopy revealed that aurasperone A and D were produced by *A*. *neoniger* [86]. When tested against a pathogenic strain of *Fusarium oxysporum*, aurasperone A and D extracts had a minimum inhibitory concentration (MIC) of 76 and 67 μg/mL, respectively [86]. Extract from an *Aspergillus* species isolated from the plant *Bethencourtia palmensis* was found to contain mellein and neoaspergillic acid, known antifungals [87]. Extracts inhibited the growth of *Alternaria alternata*, *Botrytis cinerea* and *F*. *oxysporum*, in culture, at an effective dose (mg/mL) EC50 of: (mullein) 0.44, 0.29 and 0.34, respectively, and (neoaspergillic acid) 0.01, 0.04 and 0.07, respectively [87]. 

Similarly, cultures of *Lecanicillium lecanii* and *Gliocladium catenulatum* were found to produce chitinase capable of inhibiting the growth of mycelia and conidial germination of *R*. *solani* and hyphal growth, conidial germination and sclerotial germination of *F*. *oxysporum* [88]. In another study, *Colletotrichum coccodes* and *Phyllosticta capitalensis* isolated from the Indian medicinal plant *Houttuynia cordata* were found to inhibit the growth of the opportunistic human pathogen *Candida albicans* [68]. *Colletotrichum coccodes* was found to produce geranylgeraniol (antibacterial), farnesol (anti-quorum sensing) and squalene (antioxidant, cytotoxic) [68]. *Phyllosticta capitalensis* was found to produce 1-octacosanol, an antioxidant and antibacterial compound [68]. *Diaporthe caatingaensis* isolated from the medicinal plant *Buchanania axillaris* was found to produce camptothecin, a molecule more commonly derived from plants, that has anticancer, antibacterial and antifungal properties [89,90]. Further study is required to determine if the fungal-derived camptothecin shares the same antifungal properties as the plant-derived compound.

*Trichoderma harzianum* and *T. lentiforme* were isolated from watermelon, *Citrullus lanatus*, along with 348 other fungal endophytes [91]. Seven fungal species were tested for their antagonistic abilities against 14 soil-borne pathogens: *Fusarium oxysporum* f. sp. *niveum*, *F*. *oxysporum* f. sp. *melonis*, *F. solani* f. sp. *cucurbitae*, *Macrophomina phaseolina*, *Monosporascus cannonballus*, *Neocosmospora falciformis* and *N*. *keratoplastica* [91]. *Trichoderma harzianum* and *T. lentiforme* showed the highest rates of pathogen growth inhibition of up to 93% in dual culture assays, while in vitro tests on melon and watermelon plants showed a reduction of disease occurrence of up to 67% [91]. The *Trichoderma* species were observed using several modes of action to inhibit the growth of the tested plant pathogens. These fungal endophytes outcompeted the pathogens for space and nutrients, produced compounds that degraded the cell walls of the pathogenic fungal hyphae and directly parasitized the pathogens with invading hyphae [91].

*Aspergillus terreus*, isolated as an endophyte from the seed of the rubber tree, *Hevea brasiliensis*, was found to inhibit the growth of pathogens *Rigidoporus microporus* and *Corynespora cassiicola* by 81, 64 and 70%, respectively, in dual culture assays [92]. Using a dipped stick inhibition assay, sterilized rubber tree wood inoculated with liquid culture of *A. terreus* completely inhibited the growth of *R. microsporus* [92]. Furthermore, sterilized leaves soaked in liquid culture of *A. terreus*, and then cut with a scalpel and placed onto cultures of *Corynespora cassiicola,* showed significantly reduced rates of infection of 87–93%, compared to the control [92].

However, some studies conflict on the antifungal properties of endophytic fungi. One such example involves *Diaporthe* (=*Phomopsis*), a speciose genus that includes many saprotrophs, pathogens and endophytes. *Diaporthe* sp. isolated as a stem endophyte of *Azadirachta indica* (neem) produced two 10-membered lactones with antifungal activity against several plant pathogenic fungi, including *Aspergillus niger*, *Botrytis cinerea*, *Cochliobolus heterostrophus* (=*Bipolaris maydis*), *Fusarium avenaceum*, *F. moniliforme*, *Ophiostoma minus* and *Penicillium islandicum* [44,93]. Against the aforementioned plant pathogens, 8R-acetoxymultiplolide A showed the highest antifungal activity but had little antifungal activity against *Candida albicans* [94]. Non-pathogenic *Penicillium* spp. have also shown antifungal properties in *A. indica*, but the responsible compounds remain unknown [44]. Just as fungal pathogens can exhibit narrow host preferences, fungal endophytes may exhibit similar host specificity, so the beneficial antagonistic effects observed in one host species may not be seen in another. Further study is required to elucidate the antifungal mode of action used by different fungal endophytes. This will improve the understanding of which species can be used as potential biocontrol inoculants versus producers of antifungal compounds that can be extracted and applied directly.

## 3. Fungal Endophytes and Their Activities against Bacterial Pathogens

In addition to antifungal compounds, endophytes also produce antibacterial compounds that may protect the host plant against bacterial pathogens. These antibacterial compounds vary, with some being broad spectrum but others providing protection against a narrower target group [44]. One such compound, javanicin, showed activity against many microbes, but is most effective against *Bacillus* spp. and *Escherichia coli* [44]. Other broadly antimicrobial secondary metabolites that endophytes produce include terpenoids, alkaloids, phenylpropanoids, aliphatic compounds, polyketides, acetol, hexanoic acid, acetic acid and peptides [1,95]. Phomadecalin E and 8α-acetoxyphomadecalin C are two examples of terpenoids produced by some endophytes of the genus *Microdiplodia* that show effective antibacterial properties against antagonistic strains of *Pseudomonas aeruginosa* [1]. Some strains of *Pseudomonas aeruginosa* can cause soft root rot in plants such as *Panex ginseng*, *Arabidopsis* and *Ocimum basilicum* and can also be opportunistic human pathogens [96,97].

Another fungal endophyte that produces broad-spectrum antimicrobial compounds is *Chaetomium globosum,* which exhibits activity against several pathogenic microorganisms and also has anti-biofilm activities [98]. Similarly, *Penicillium* sp. isolated from the medicinal plant *Stephania dielsiana* shows remarkable broad-spectrum antimicrobial activity, with the MIC of the EtOAc extract ranging from 1.2 to 6 mg/mL against seven different animal pathogenic bacteria [99]. The crude extract of *Trichoderma harzianum*, a fungal endophyte isolated from *Rosmarinus officinalis*, showed significant antimicrobial activity against *P. aeruginosa*, *Staphylococcus*
*aureus*, *Klebsiella pneumoniae*, *B. subtilis* and *E. coli*, which suggests that this endophyte also has a potential to be used as biocontrol agent against phytopathogenic bacteria [100]. *Diaporthe phaseolorum*, *Aspergillus fumigatus* and *A. versicolor*, isolated as endophytes from healthy tomato (*Solanum lycopersicum*) plants, produced antibacterial metabolites such as acetol, hexanoic acid and acetic acid, which showed effective biocontrol activities against bacterial spot of tomato (*Xanthomonas vesicatoria*) [95]. Extracts containing extracellular metabolites of endophytic *Aspergillus* spp. from *Cupressaceae* hosts showed varying antibacterial effects against *Bacillus* sp., *Erwinia amylovora* and *Pseudomonas syringae*, although the metabolites were not identified [101]. Secondary metabolites that are effective against multiple pathogens, such as cycloepoxylactone, are especially useful in plant defence [1]. These antimicrobial metabolites can be directly produced by an endophytic fungus or can be produced by the host plant in response to endophyte inoculation [1]. Knowledge of the secretion of these compounds and associated gene expression remains limited [1].

## 4. Fungal Endophytes and Their Effects against Plant-Parasitic Nematodes

Plant-parasitic nematodes (PPN) are a major threat to agricultural crops worldwide, causing $215.8 billion USD worth of damage in the USA alone, and are of particular concern in tropical and subtropical regions [1,102,103]. Nematodes form feeding sites on plant roots and stems, from which nutrients are extracted, which creates wounds through which secondary opportunistic fungal, bacterial or viral pathogens can enter the plant [104]. They also serve as vectors for viruses that may infect crop plants and cause disease or death in host plants [1]. Traditionally, chemical-based nematicides are used to inhibit the presence and spread of nematodes. However, the chemical applications can have non-target effects like other pesticides, which damage the assemblage of beneficial microbial communities in the rhizosphere and surrounding soil [105]. As such, there is a growing interest in finding microorganisms that may co-exist in the soil or plant tissues and can inhibit the growth and spread of nematodes [106]. Several fungal endophytes have been reported that either produce nematocidal compounds, parasitize nematode eggs and larvae or utilize hyphal loops and other means to trap nematodes and their eggs [106]. Some fungal species appear to produce bioactive compounds that directly or indirectly impact nematode colonization of the plant and/or surrounding soil, but the exact chemical compounds responsible for these effects are still being elucidated [106].

Root-knot nematodes, represented by *Meloidogyne* species, are globally ubiquitous and impact over 2000 plant species including economically important crops such as tomato, cotton, cucumber, melon, soybean and rice [104,106,107,108,109,110,111,112]. Many fungal genera have been reported as having inhibitory effects on *Meloidogyne* species, including: *Acremonium*, *Alternaria*, *Arthrobotrys*, *Chaetomium*, *Cladosporium*, *Clonostachys*, *Diaporthe*, *Drechslerella*, *Epichloë*, *Epiccocum*, *Fusarium*, *Gibellulopsis*, *Melanconium*, *Metacordyceps*, *Monacrosporium*, *Neotyphodium*, *Paecilomyces*, *Phialemonium*, *Phyllosticta*, *Piriformospora*, *Purpureocillium*, *Talaromyces* and *Trichoderma* [106,110,111,112,113,114,115,116,117,118,119,120,121,122,123,124,125,126]. Species from one or more of these genera have also been reported as having similar antagonistic effects towards other species of nematodes [106]. The presence of one or more species has been reported as significantly decreasing the occurrence of root knots and the nematodes that cause them.

Compounds produced by *Alternaria*, *Chaetomium*, *Cladosporium*, *Clonostachys Fusarium*, *Phyllosticta*, *Piriformospora* and *Trichoderma* strains have been shown to alter the chemical composition of existing metabolites, or increase their production, within the host plant resulting in plant growth promotion or induced resistance to invading nematodes [112,113,120,124,127,128]. Alternatively, *Acremonium*, *Diaporthe*, *Epichloë*, *Melanconium*, *Phialemonium* and *Purpureocillium* species can produce bioactive compounds that directly inhibit nematode eggs, juveniles, and females [108,115,121,124,128,129]. Strains of *Chaetomium*, *Clonostachys*, *Phyllosticta* and *Trichoderma* have also been reported as hyper colonizers that can outcompete plant pathogens, including nematodes, for space and nutrients within the plant host [110,112,113,123,124].

*Fusarium* species are the most commonly reported fungi known to have antagonistic effects on nematodes through the production of bioactive compounds that improve plant growth and induce systemic resistance to nematodes, or directly inhibit the growth and development of nematodes [112,114,119,127,128]. *Fusarium* species were shown to alter the production of growth hormones, as well as the composition of root exudates, produced by the host plant, subsequently decreasing colonization by *M*. *incognita* [112]. *Fusarium oxysporum* was shown to induce plant resistance to *M*. *incognita* by triggering the production of unknown compounds by the host plant [120]. Similarly, banana plants inoculated with *Fusarium* sp. showed reduced parasitism by the burrowing nematode *Radopholus similis* due to induced systemic resistance (ISR) [130]. More recent work with *F*. *oxysporum* strain 162 identified 11 compounds, nine of which had some nematocidal effect; 4-hydroxybenzoic acid, indole-3-acetic acid (IAA) and gibepyrone D were the most effective, with a lethal dose of 50% of the test organisms (LD 50) concentration of 104, 117 and 134 µg/mL, respectively, after 72 h [127]. The production of IAA suggests that this compound serves a dual function by improving plant health and resistance to nematodes while also being secreted as toxin [127].

A study examining the mechanism of action found that within 10 min of exposure to *F. oxysporum* nematocidal compounds nematode motility decreased, and within 24 h exposed nematodes were dead [131]. The compounds were most effective against sedentary nematodes compared to migratory nematodes, with non-parasitic nematodes remaining unaffected regardless of their mobility [131]. The observed effects of *F. oxysporum* on target versus non-target nematodes is important because it reduces populations of plant pathogenic species without harming non-pathogenic nematodes that may feed on pathogenic bacteria and fungi or parasitize crop pests [132]. In a recent study, several fungal endophytes belonging to the genera *Alternaria*, *Chaetomium*, *Cladosporium*, *Diaporthe*, *Epicoccum*, *Gibellulopsis* and *Purpureocillium* isolated from cotton plants were successfully used as a seed treatment that reduced damage caused by *Meloidogyne incognita* [124]. In another study, sacha inchi (*Plukenetia volubilis*) plants inoculated with *Trichoderma* and *Clonostachys* significantly reduced the damage and number of galls induced by root-knot nematodes compared to non-inoculated plants [123].

The genus *Epichloë* contains endophytic fungi best known for forming mutual symbioses with a variety of grass species [129,133,134]. Members of this genus colonize grass tissues through hyphal expansion, though this is most prevalent in shoot material [133]. *Epichloë* spp. are well known for their ability to produce bioprotective alkaloid and other non-alkaloid secondary metabolites [133,134,135]. *Epichloë coenophialum* has been reported to significantly decrease parasitism by *M*. *marylandi* and migratory root lesion nematodes from the genus *Pratylenchus*. Although the exact mode of action remains uncertain, it is likely in part from alkaloid production [133]. However, as this fungus is typically not present in the plant roots, these compounds must be translocated from shoot to root by the plant, or the compounds induce resistance [133].

A strain of *Chaetomium globosum* was found to produce 1,2-benzenedicarboxaldehyde-3,4,5-trihydroxy-6-methyl, also known as flavipin, which is a potent antioxidant and antagonist of nematodes [118]. *Purpureocillium lilacinum* produces proteases and chintinases, which interfere with the successful development of nematode eggs of both *Meloidogyne* and *Heterodera* species; *P. lilacinum* is also known to parasitize eggs through hyphal penetration [124,136]. *Diaporthe phaseolorum* (=*Phomopsis phaseoli*) and *Melanconium betulinum* were found to produce 3-hydroxypropionic acid which showed selective nematocidal capacity with anLD50 concentrations of 12.5–15 µg/mL when applied to *M*. *incognita* [121]. Three chlorinated, epimeric oxazinane derivatives isolated from *Geotrichum* sp. showed nematocidal activity against the nematode species *Bursaphelenchus xylophilus* and *Panagrellus redivivus* [44,137]. Species from the genera *Dactylonectria*, *Epicoccum*, *Fusarium* and *Myrothecium* were all found to produce bioactive compounds with high activity against second-stage juveniles of *H*. *glycines* [111].

Endophyte strains of *Fusarium solani* and *Acremonium implicatum* were reported as direct parasites of eggs, juveniles and females from the genus *Meloidogyne* [115,122,128]. Parasitism by these fungi occurs through hyphal extension and penetration of the nematode cellular structures [115,122,128]. Meanwhile, known nematode parasites *Arthrobotrys iridis*, *Metacordyceps chlamydosporia* and *Hirsutella rhossiliensis*, thought to originate in the soil, have been found occurring as plant root endophytes [111]. These fungi are best known for switching from a saprophytic to a parasitic lifestyle when exposed to nematodes [111,138]. Hyphal structures such as loops and nets, and paralyzing secretions, are used to trap nematodes before hyphae penetrate the cuticle and colonize the body [138,139]. *Metacordyceps chlamydosporia* produces an alkaline serine protease, which digests the outer membrane of nematode eggs, allowing for hyphal penetration and infection of the eggs of both *Meloidogyne* and *Heterodera* species [111,140]. *Hirsutella* spp. are parasitic to nematodes of both sedentary and migratory lifestyles, including *Ditylenchus*, *Heterodera*, *Meloidogyne*, *Pratylenchus* and *Rotylenchus* [136]. *Hirsutella rhossiliensis* produces sticky conidia that attach to the cuticle of a nematode upon contact. The conidium then produces a germination tube that penetrates the cuticle, and hyphae rapidly colonize and kill the nematode [136]. In the forestry sector, the nematophagous endophyte *Esteya vermicola* shows promise as a biocontrol agent of the invasive pinewood nematode (*Bursaphelenchus xylophilus*), with studies showing inoculation with *E. vermicola* significantly increases survival rates following *B. xylophilus* inoculation [141,142,143].

Endophytic fungi have been used as a seed treatment of agricultural plants for the control of nematodes, with other practices involving root inoculation [112,119]. The full capacity of fungal endophytes as nematode control agents remains understudied but shows potential for the development of effective biocontrol methods. The elucidation of the bioactive compounds produced by the endophytes, or whose production is induced within the host plant to combat nematodes will aid in understanding the mode of action for these compounds and how they directly or indirectly inhibit nematode development. Fungal species known to parasitize nematodes need to be further investigated for their abilities against different genera of nematodes for their use as biocontrol agents.

## 5. The Effect of Fungal Endophytes against Plant Viral Diseases

Fungal endophytes reduce viral diseases either by increasing plant defences or by reducing the spread of viruses by having entomopathogenic activities against vectors that spread the viruses. Although it is not within the scope of this review, it has been shown that several fungal endophytes have anticancer and antiviral properties against human viruses [144,145,146,147]. Studies investigating the antiviral properties of fungal endophytes against plant viruses involve foliar inoculation of viruses on endophyte inoculated plants, although few such studies exist [148]. Inoculation of *Lolium pratense* (meadow ryegrass) with *Neotyphodium uncinatum* reduced viral infection of *Barley yellow dwarf virus* in inoculated plants, likely due to the production of alkaloids that deterred viruliferous aphid vectors and indirectly reduced the spread of virus infection [149]. In another study, inoculation of squash plants with different strains of *Beauveria bassiana* provided protection against *Zucchini yellow mosaic virus* compared to the non-inoculated control plants [148]. The antiviral defence of fungal endophytes may be specific against different viruses infecting the same plant species. Maize plants inoculated with *Trichoderma harzianum* and *Metarhizium anisopliae* were more resistant to *Sugarcane mosaic virus* compared to the control plants, while the same inoculated plants were not significantly resistant to *Maize chlorotic mottle virus* [150]. Environmental conditions also play a role in endophyte-induced plant resistance against plant viruses. Inoculation of tomato plants with *Piriformospora indica* repressed the amount of *Pepino mosaic virus* in shoots under higher light intensities, while significantly increasing fruit biomass [151]. In general, the most prevalent way to protect against viral infection of plants is by attempting to limit the potential viral vectors prior to infection [152]. Typically, this process involves the use of insecticides or other potentially harmful compounds for control [152]. Endophytic priming of plants represents a potential treatment option that could reduce the application of insecticides and may also provide persistent protection if insecticidal treatments fail [152].

## 6. The Role of Fungal Endophytes against Mites

Phytophagous mites are globally important pests of agricultural crops and ornamental plants, causing damage through feeding and by transmitting viruses and subsequently reducing photosynthetic capacity, overall health, yield, and market value. Mite pests can have exceptionally broad host ranges; for example, the two-spotted spider mite (*Tetranychus urticae*; TSSM) is reported from at least 1000 plant species across 130 families, on which it can cause significant yield losses in commercially important crops such as cucurbits, beans, hops, grapes, apples and strawberry [153,154]. The most economically important mite species include spider mites (*Tetranychus* spp.), the citrus red mite (*Panonychus citri*) and the European red mite (*P. ulmi*). Control generally involves the application of acaricides and biological control using natural predators. The 2013 acaricide market was estimated to be worth approximately €900 million, not including broad-spectrum pesticides also applied for mite control, and in 2008 approximately 80% of the total market value was spent on the control of spider mites alone [155]. Pesticide resistance in phytophagous mites is a serious issue to agroecosystems; for example, TSSM and European red mite are among the most resistant species, with the former showing 400+ cases of resistance across 90+ compounds and the latter showing almost 200 reported cases of resistance across almost 50 compounds [156]. Careful and strategic application is required to reduce multiple acaricide resistance and to reduce effects on non-target natural enemies (e.g., predatory mites) in integrated pest management systems [157,158,159,160].

Another control tool involves the application of mycoacaricides, which include well-known entomopathogenic hypocrealean fungi such as *Akanthomyces muscarius*, *Beauveria bassiana*, *Cordyceps fumosorosea*, *Hirsutella thompsonii*, *Metarhizium anisopliae* and *Purpureocillium lilacinum* [161,162,163,164]. These generalist entomopathogens typically infect insects via conidia, which land on the insect cuticle, germinate, and form an appressorium that penetrates the cuticle through a combination of mechanical pressure and cuticle-degrading enzymes [165]. The fungus then proliferates throughout the insect hemolymph via yeast-like hyphal bodies or blastospores, colonizes internal tissues and may produce toxic secondary metabolites. Dead insects appear mummified and are the source for new infective propagules.

Hypocrealean entomopathogens/acaripathogens are well-studied and have been extensively reviewed, primarily as biocontrol agents of insects but also of mites [6,164,166,167,168,169,170,171,172,173]. *Beauveria* and *Metarhizhium* are by far the most studied mycoacaricides and mycoinsecticides and can endophytically colonize a broad range of host plants naturally and when applied by methods such as seed soaking and coating, root dip, foliar spray, wound inoculation and soil treatment [171]. Interestingly, entomopathogenic endophytes can be recovered from both root and foliar tissues following seed inoculation, suggesting systemic acropetal growth, which offers a convenient and effective method of application [174,175]. For example, foliar endophyte colonization was confirmed in cotton seeds (*Gossypium hirsutum*) that were soaked in conidia suspensions of either *Beauveria bassiana* or *Purpureocillium lilacinum*, both of which subsequently reduced cotton aphid (*Aphis gossypii*) reproduction in field trials [176]. Composted cabbage waste (*Brassica oleracea* var. *capitata*) inoculated with *Clonostachys rosea* and used as a medium to cultivate tomatoes resulted in a 100% endophyte colonization rate; however, the endophyte colonization did not significantly decrease populations of TSSM [177]. A tomato leaf detachment bioassay with *B. bassiana*-inoculated plants showed significant increases in mortality of TSSM depending on the inoculation method; mortality and leaf endophyte colonization frequency were mutually highest in sprayed leaves followed by soil drenching and seed soaking [178]. Commercial strains of *Trichoderma asperellum*, *T. atroviride* and *Cordyceps fumosorosea* applied as soil drench significantly reduced the number of TSSM and green peach aphid (*Myzus persicae*) on pepper (*Capsicum annuum*) [179].

Tomato seedlings inoculated with a strain of *Fusarium solani* isolated from tomato roots significantly reduced the number of TSSM eggs compared to untreated control plants, but this did not affect the number of live adult females found alive [180]. In spider mite-infested plants colonized by *F. solani*, JA and SA defence marker genes were up-regulated and volatile emissions were altered and more attractant to *Macrolophus pygmaeus*, a natural predator of spider mites. The protectant activities of endophytes against mites can therefore involve antibiosis, feeding deterrence and defence priming, including attracting natural predators.

The yeast-like basidiomycete *Meira geulakonigii*, originally isolated from citrus rust mite (*Phyllocoptruta oleivora*) cadavers on grapefruit (*Citrus paradisi*) in Israel, was later reported as an endophyte of fruit peels of grapefruit [181,182]. *Meira geulakonigii* causes significant mortality of the citrus rust mite and other mites, possibly due to the secretion of toxic metabolites [181,183]. In another study, *M. geulakonigii* resulted in an almost 100% mortality of citrus rust mites, 80% mortality of citrus red mites and carmine spider mites (*Tetranychus cinnabarinus*) and a significant reduction of powdery mildew (*Podosphaera fusca*) when sprayed on cucumber leaves [184]. *Meira argovae* produces argovin (4,5-dihydroxyindan-1-one), which was observed to kill 100% of citrus rust mite populations at 0.2 mg/mL [185]. While *M. argovae* was first isolated from cadavers of carmine spider mites on leaves of castor bean (*Ricinus communis*) in Israel, it was later isolated from young shoot tissues of bamboo (*Phyllostachys bambusoides*) with witches’ broom disease (*Aciculosporium take*) in Japan [182,186]. The genus *Meria* includes species isolated from Japanese pear (*Pyrus pyrifolia*) fruits, rhizosphere soil of tobacco roots, the surfaces of *Magnolia* leaves and vetiver grass (*Chrysopogon zizanioides*) leaves and as an endophyte of *Abies beshanzuensis* [187,188,189,190,191]. The identification of acaropathogenic *Meira* species and their overall association with the rhizosphere, phyllosphere and endosphere of plants suggests their potential application for controlling phytophagous mites.

Hypocrealean entomopathogens/acaripathogens are the most promising insect and mite biocontrol fungi. Species with endophytic life histories may be particularly useful as they can be conveniently applied (e.g., via seed soaking or coating), persist and spread within the host crop plant, prime host defence pathways and offer protection against a broad range of pests (not limited to mites) and may be less susceptible to factors limiting efficacy in the field (low moisture and UV light) [171,192,193,194]. Furthermore, evidence suggests that some acaripathogens may be compatible with predatory mites and, in some cases, can have a synergistic effect [195,196,197,198,199,200,201,202,203] although negative interactions are reported [204,205,206,207]. Endophytic mycoacarcides may therefore play an increasingly important role in future integrated pest management systems to control phytophagous mites and reduce acaricide resistance [208].

## 7. Environmental Factors Affecting Endophytic Fungi and Plants

The symbiosis between endophyte and plantcan be affected by various environmental factors [209]. Weather is among the top factors and can influence the frequency of endophyte occurrence [209]. For example, wind is a primary spore dispersal mechanism for endophytes and, therefore, dispersal would be increased in areas of higher winds [210]. Similarly, increased precipitation is also linked to enhanced prevalence of endophytes, specifically those that are transmitted horizontally due, in part, to spore dispersal [209,210]. Along with dispersal, these endophytes rely on moisture to germinate and colonize the host plant. Factors such as temperature and solar radiation can make environments either welcoming or inhospitable to endophytes, which generally only survive in specific temperature ranges [211].

Data suggest that the diversity and colonization rate of endophytes is not static [212]. Seasonal changes, specifically in the spring, have shown higher colonization rates and diversity than in the fall [212]. These data are complicated by the previously discussed environmental factors associated with season, but season can be used to generalize those environmental factors [211]. The location and age of plants can have an effect on the endophyte density as well, with older leaves having stronger resistance to colonization than younger leaves [213,214]. Surprisingly, both leaf chemistry and toughness have not been shown to significantly change colonization [214].

Data exists on the ability of endophytes to enhance their host plant ability to tolerate stressors such as salinity, drought, and other extreme weather events [49]. Stress tolerance may be increased due to antioxidant compounds such as phenolic acids, isobenzofuranones, isobenzofurans, mannitol and other carbohydrates [71]. Endophytes may produce antioxidants, and they have also been shown to release reactive oxygen species to stimulate the host plant to produce such antioxidants [71]. These low-weight antioxidants interact with several plant cellular components and modulate processes such as mitosis and cell elongation, as well as senescence and apoptosis, to influence plant growth and development [215].

## 8. Host Plant Feedback on Endophytes

Generally mutualistic, the symbioses between endophytes and plants provide the endophyte with protection from abiotic and biotic stress and enhanced competitive abilities, while the plant receives protection and in some cases nutrients [49,216]. This mutual feedback is often essential for the survival of both partners [71]. However, endophytes may turn pathogenic due to nutrient shortages or prolonged severe weather [216]. A fungal species may be endophytic in one host species and pathogenic to another, so endophytic status cannot be assumed [71]. These co-evolved interactions are plastic and can be expected to destabilize under severe climate change scenarios [217,218].

Secondary metabolites can be made by either the endophyte or plant [219]. They give plants control in the relationship, allowing them to limit endophytic growth within their tissues by using lignin and other cell wall deposits to restrict or allow further colonization [71,219]. This process is also crucial for initiating the relationship and allowing colonization. Endophytes must bypass plant defence mechanisms to initially colonize the plant [40]. When plants sense an invader, they have numerous defences to try to thwart the attempt. These defence signaling cascades are initiated from recognition of fungal invasion and damage to plant tissue and may include cell wall thickening and production of secondary metabolites [220]. Host plants may also manipulate the secondary metabolites produced by endophytes to give them increased benefits for certain stressors, allowing the plant to adjust what is needed and when [219]. They may also modify the metabolites if they are too toxic and are causing harm to the plant [219].

## 9. Endophyte Transmission

The transmission of endophytes can occur vertically, with the parent plant passing on endophytes to their offspring through seeds. In this manner, the endophyte is present for the entire plant life cycle [221]. Vertical transmission is most common among grass species, which may only have one endophyte species and have only a single genotype for that endophyte [209]. Endophytes are also transmitted horizontally, often by spores present in the surrounding environment [221,222]. Seedlings may begin their lives free of endophytic colonization and gradually become colonized, with an accumulation at the end of the growing season, by spores from rain, air or passing organisms such as insects or mammals [210]. This mode of transmission provides a heterogeneous endophytic community that is different from that of the parent plant and may lead to more resilient populations [209]. Modulating crop plant microbiomes can incorporate both vertical and horizontal transmission; for example, inoculating maternal plants with endophytes of interest to establish endophyte-colonized seeds and applying endophyte inocula via seed coat treatments, growth media amendments or aerial sprays [223,224]. Studying the transmission and life histories of endophytes will therefore provide practical knowledge that can be applied to developing more effective inoculants and application techniques.

## 10. Final Thoughts

As the plant health paradigm continues to shift into a more holistic view incorporating both the plant and its microbiome, i.e., the holobiont, the promise of improving plant productivity, health and resiliency by improving resistance to pathogens and pests through microbiome manipulation becomes more enticing. Plant microbiomes may be more precisely engineered and customized by inoculating with specific endophytes or endophyte consortia (Figure 1). Strategies for selecting and applying target endophytes can be preemptive, for example, considering protection and beneficial traits in anticipation of expected fungal pathogens, or reactive, for example the rapid application of inoculants to mitigate an acute health issue such as a mite outbreak [225]. Rather than selecting individual endophytes or consortia, a broader approach can involve microbiome transplantation or the use of soil amendments and root exudates to attract and maintain beneficial microbiomes [226,227,228]. A natural extension of modulating the host microbiome is to consider and optimize the plant’s interactions and receptivity through microbiome breeding programs [227].

As this is an emerging field, much research is required before microbiome engineering shows predictable and consistent benefits that will lead to its widespread adoption. Immediate research priorities include not just identifying beneficial endophytes and other microbial symbionts and elucidating their modes of action, but also measuring and assessing inoculant establishment, conducting longer-term studies of temporal dynamics of functional changes following interventions and identifying barriers to establishment. Given the complex interactions between hosts and endophytes, which can include switching from mutualist to opportunist pathogen, candidate endophytes should be studied in planta under different conditions, including abiotic stress. Agricultural applications of endophytes also present some health and safety considerations; for example, if the endophyte is capable of colonizing plant tissues intended for consumption. Endophytes and epiphytes have been applied to combat postharvest diseases in apples, bananas, citrus, grapes and other fruits, and future considerations may include beneficial effects on the human gut microbiome [229,230,231,232,233].

These research directions provide exciting and open opportunities not just to answer fundamental questions required for the application of fungal endophytes in pest and disease management, but also to contribute to our knowledge of fungal biodiversity, fungal and plant ecology and complex multipartite interactions. A greater understanding of microbiomes and plant health will also provide novel monitoring solutions for predicting future disease outcomes linked to pathobiomes and dysbiosis. The future of agriculture will involve the increasing consideration and integration of the plant microbiome in pest and disease management strategies and, as crucial members of the plant microbiome, fungal endophytes will play a leading role.

Although there are many different endophytes of documented benefit to agricultural plants, we have compiled an annotated list of some of the most important (Table 1).

## Figures and Tables

**Figure 1 plants-11-00384-f001:**
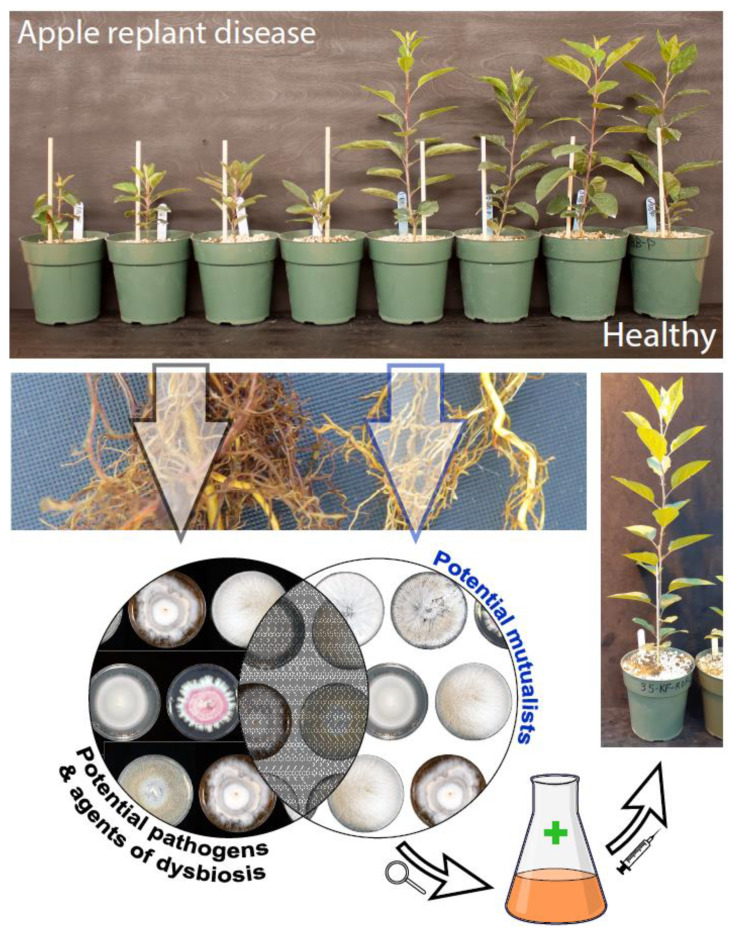
A simplified flow diagram illustrating an approach to isolating, identifying and developing plant health promoting endophytes. This example involves apple replant disease (ARD), a complex phenomenon describing the detrimental physiological and morphological reaction of apple trees in sites that have been repeatedly planted with apple, which can cause a shift towards soil microbiome dysbiosis. The four apple plants on the left show stunting from being grown in pots containing ARD soil, and the four apple plants on the right show good growth and health from being grown in non-ARD soil. Root endophyte and rhizosphere diversity of ARD and non-ARD soils and plants is characterized and compared using a combined approach involving metabarcoding and culturing. Potential pathogens, causal agents and indicators of dysbiosis are identified in ARD samples, and potential mutualists are inferred from non-ARD samples. Mutualistic individuals or consortia are selected as possible ARD biocontrol agents and investigated for their antagonism against ARD-associated pathogens, protective secondary metabolites, plant health promoting interactions and other attributes. Beneficial rhizosphere strains and/or endophyte(s) are then inoculated into soil and/or apple plants for further study, including challenge trials involving inoculated apple planted in ARD soils.

**Table 1 plants-11-00384-t001:** Annotated list of selected important plants and the endophytes they harbor. Note: * = see article, ** = see table in article, N/A; not applicable.

Endophyte	Host Plant	Infection Location	Research Topic	Notable Findings	Reference
Numerous **	Azadirachta indica	Numerous **	Discusses the antimicrobial, antioxidant and pathogenicity target compounds produced by the endophytic fungi.	N/A	[44]
Numerous **	Dendrobium moniliforme	Roots	Identifying the endophytic fungi and their role in plant growth and development.	Nine fungi isolated; unidentified Fusarium sp. was dominant. The presence of phenolic compounds suggests their contribution to antimicrobial and antioxidant properties for their host plant. Colletotrichum alatae showed highest concentration of IAA and as a fungal elicitor it resulted in the highest total chlorophyll content.	[234]
Numerous *	Dendrobium loddigesii	Roots and seeds	The diversity of endophytic fungi was explored and cultures were tested for antimicrobial activity.	Forty-eight isolates identified to 18 genera including Fusarium and Acremonium. Antimicrobial activity was tested on 17 isolates belonging to 9 genera and again Fusarium was dominant.	[235]
Many Fusarium spp. **	Orchid spp. **		Fusarium-orchid interactions and the challenges when dealing with the pathogen.	There is evidence that Fusarium can induce host resistance against many pathogens in crops such as banana, tomato, as well as orchid.	[236]
Trichoderma spp.	Numerous **		Overview of Trichoderma spp. as symbionts.	Many Trichoderma spp., including T. virens, T. atroviride and T. harzianum can induce localized and systemic host plant resistance to a variety of plant pathogens. Induced resistance increases the expression of defence-related genes in the plant, similar to systematic acquired resistance. Generally, this is short term, except for in one case (T. asperellum and cucumber) where a longer response was shown, and elements were similar to rhizobacteria-induced systemic resistance.	[237]
Clavicipitaceae and others **	Grasses **		Overview of endophytic fungi in grasses.	Protection against plant pathogens is a possible benefit as seen in endophyte infected tall fescue being resistant to seedling blight (a disease caused by Rhizoctonia). Infected plants are also more resistant to oat crown rust (Puccinia coronata) compared to uninfected plants. Tall fescue was more resistant to barley yellow dwarf virus, with uninfected plants showing twice the frequency of disease. This shows deterrence of aphid vectors of the virus. Panicum agrostoides (a wetland grass) had less leaf blight (Alternaria triticina) infection when infected with Balansia henningsiana. Epichlöe- infected timothy grass was resistant to purple eyespot disease (Cladosporium phlei)	[238]
Trichoderma reesei, T. atroviride and T. virens.	N/A	N/A	Identifying gene clusters associated with secondary metabolism in Trichoderma spp.	One new NRPS and six new PKS clusters were found in the Trichoderma reesei genome. T. atroviride had four NRPS and eight PKS clusters while T. virens had four NRPS and 8 PKS clusters.	[239]
Trichoderma spp. *	N/A	N/A	Discussing the bioactivity, regulation and biological roles of secondary metabolites produced by Trichoderma spp.		[240]
Trichoderma atroviride, T. reesei and T. virens	N/A	N/A	Looking at the mechanisms of mycoparasitism by comparing the transcriptional responses of Trichoderma spp. with different lifestyles against Rhizoctonia solani.	Trichoderma atroviride and T. virens expressed different genes for antagonism when confronted with R. solani. T. virens up-regulated genes for gliotoxin biosynthesis, poisoning R. solani, while T. atroviride followed a strategy involving antibiosis and hydrolytic enzymes. T. reesei appeared to mainly express genes for nutrient acquisition suggesting an attempt at competition instead of mycoparasitism.	[241]
Trichoderma atroviride, T. reesei and T. virens	N/A	N/A	Comparing genomes of different Trichoderma spp.	Genome analysis and comparison of Trichoderma atroviride, T. virens and T. reesei. Phylogenetic analysis showed that T. reesei and T. virens derived from T. atroviride, suggesting mycoparasitism-specific genes arose in a common Trichoderma ancestor but were lost in T. reesei.	[242]
Fusarium equiseti, Pochonia chlamydosporia	Barley	Roots	Evaluating the root population dynamics of fungi under non-axenic conditions. Fungi were examined for their presence, effect on plant growth and response to Gaeumannomyces graminis var. tritici (causal agent of take-all disease).	Both fungi can protect host plants from G. graminis var. tritici in laboratory conditions. Clear suppressive effect on the pathogen could not be detected but F. equiseti isolates reduced the mean root lesion length. Root colonization by P. chlamydosporia promoted plant growth.	[128]
Many including Cryptosporiopsis cf. quercina, Colletotrichum spp.	N/A	N/A	Brief review of biological activities and applications of endophytes.	Suggest that the nutritional status and fitness of the host plant (which are enhanced by the endophytes) as well as their ability to tolerate abiotic stress are key factors in the plants ability to resist disease. Cryptosporiopsis cf. quercina and Colletotrichum spp. have been shown to be effective against plant pathogens including Rhizoctonia cerealis, Phytophthora capsici, Pyricularia oryzae and Gaeumannomyces graminis. Endophytes demonstrate potential for phytoremediation.	[51]
97 isolates **	12 genera of orchids	Leaves, stems, flowers	Analysing the antifungal, antioxidant, chemical composition and antimutagenicity properties of compounds produced by fungal endophytes.	Thirteen endophyte isolates showed antifungal activity against Fusarium sp., Colletotrichum sp. and Curvularia sp. Fusarium oxysporum strain showed the highest antifungal activity and was selected for further study including characterizing secondary metabolites.	[243]
Numerous **	Stanhopea tigrine	Leaf, pseudobulb, root and flower	Examining the microbiome of Stanhopea tigrine.	Used morphological and molecular characteristics for identification and found 63 genera, with Trichoderma, Penicillium, Fusarium and Aspergillus as the dominant genera. 21 fungal isolates produced gibberellins.	[244]
Numerous **	Cephalanthera longibracteata	Roots	The goal was to determine if the fungal communities were preferentially correlated with the sites.	Thirty species of fungi were identified, endophytic community composition was affected by site.	[245]
Numerous **	Dendrobium nobile, Dendrobium chrysanthum	Mature roots and protocorms	Analyzing diversity of fungal symbionts of threatened plant species to improve conservation and commercial production.	A total of 127 fungi were isolated: Xylaria, Fusarium, Trichoderma, Colletotrichum, Pestalotiopsis, and Diaporthe were dominant.	[246]
Numerous **	Cyrtochilum myanthum, Scaphyglottis punctulata, Stelis superbiens	Roots	Analyzing the diversity of fungal root associates for conservation purposes.	A total of 115 fungal isolates were identified corresponding to 49 OTUs. Ascomycetes were dominant, with Trichoderma sp. as the most frequent taxon.	[247]
Numerous **	Pomatocalpa decipiens	Leaf segments and root	Obtaining potential phosphate solubilising strains from endophytic mycoflora.	A total of 928 endophytic phosphate solubilising fungal isolates were obtained from the leaf segments. Twenty endophytic phosphate solubilising fungi were isolated from the root samples.	[248]
Numerous including saprotrophic basidiomycetes *	Mycoheterotrophy orchids		Investigating how Mycoheterotrophic orchids receive their carbon in regions where ectomycorrhizal fungi, are not present.	Different fungi were found and identified. Research suggests that temperature and moisture in rainforests may favour sufficient saprotrophic activity to support development of mycoheterotrophy.	[249]
Numerous	N/A	N/A	What makes a fungus parasitic or endophytic and how plants avoid exploitation by parasites but benefit from mutualistic endophytes.	If the symbiosis is not equal, disease symptoms appear on the host plant and/or the fungus is expelled by host defence reactions and no longer receives benefits.	[48]
Numerous *	Heisteria concinna, Ouratea lucens	Leaves	Endophyte colonization patterns, richness, host preference and spatial variation were examined.	A total of 347 taxa were collected. Host preference and spatial heterogeneity were suggested by the data.	[46]
Numerous *	Sasa borealis, Potentilla fragarioides, Viola mandshurica	Leaves	Looking at the effects of foliar endophytic fungi and AMF on community structure in experimental microcosms.	Endophytic fungi were isolated and identified to species level. Results of this study show that AMF affect plant productivity and plant community structure.	[250]
Numerous *	Camptotheca cuminata, Gastrodia elata, Pinellia ternate	Leaves, twigs, root tissues, flower tissues	Looking at potential sources for biomedical compounds.	A total of 193 endophytes were isolated and 42 taxa were identified and tested for different bioactive compounds. Analagous bioactive compounds were produced in host endophyte cultures: three taxa isolated from C. cuminata produced high yields of camptothecin, Colletotrichum gloeosporioides from C. cuminata produced 10-hydroxycamptothecin, three taxa isolated from G. elata produced gastrodin, three taxa from P. ternata produced low amounts of ephedrine hydrochloride.	[251]
Neotyphodium coenophialum	Festuca arundinacea	Root	Greenhouse experiment conducted to identify effects of endophyte strains on copper acquisition by tall fescue varieties.	Extracellular root exudates of infected plants had a higher copper binding activity.	[252]
Numerous *	Gymnadenia conopsea	Root	Looking at the different factors that determine the spatial structure and presence of fungi associated with orchid roots.	The investigation revealed a large diversity and taxonomical range of fungi. This diversity is likely responsible for the orchids ability to live in such diverse habitats.	[253]
Numerous *	Laelia autumnalis, L. speciosa, Euchile citrina, P. squalida	Root	Looking at the community composition and diversity of fungi associated with orchids.	A total of 71 isolates were obtained, representing 20 genera. Euchile citrina showed the lowest endophytic diversity implying that the plant is specific when choosing endophytes. L. speciosa and P. squalida were generalists.	[254]
Numerous including Epulorhiza spp. and Tulasnella spp. *	Paphiopedilum, Cymbidium, Dendrobium.	Root	Looking at the diversity of fungi in orchids in understudied sites.	Twenty-seven fungal isolates were identified including Epulorhiza repens (the most common fungi found in roots from all three genera) and Epulorhiza calendulina (only found in Paphiopedilum species). Four new Tulasnella spp. were isolated and described.	[255]
Numerous *	Dendrobium sinense	Roots	Analyzing whether the endophytes were preferentially correlated with the host tree species.	A total of 56 fungal species were identified and results show that species richness and diversity were influenced by host tree species. D. sinense roots had the highest diversity.	[256]
Numerous **	Aerides odorata, Arundina graminifolia, Cymbidium aloifolium, Cymbidium munronianum, Dendrobium fimbriatum, Dendrobium moschatum, Eria flava, Paphiopedilum fairrieanum, Pholidota imbricata, Rhynchostylis retusa, Vanilla planifolia	Leaf and root tissues	Analyzing endophyte assemblages.	Xylaria spp. were found in both the leaves and the roots. The diversity of endophytes was higher in the leaves and tissue specificity was shown.	[41]
Pestalotiopsis versicolor and P. neglecta	Taxus cuspidata	Healthy leaves and bark	Investigating alternative sources of taxol.	The fungi screened produced taxol and showed a strong cytotoxic activity in the in vitro culture of tested human cancer cells.	[257]

## Data Availability

The data presented in this study are available within the article.

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
