# Peer review of "Fungal Endophytes and Their Role in Agricultural Plant Protection against Pests and Pathogens"

_plants, 2022, doi:10.3390/plants11030384_

Round 1

Reviewer 1 Report

This review is quite extensive and with a clear common thread. I believe that it will be very useful for readers who are starting in the field of plant-microorganism interaction and the state of the art of role of endophytes.

An interesting topic is the endophytic fungi genetic regulation inside and outside the host. The authors could also address the host transcriptional response when living with endophytic fungi.

Author Response

Reviewer 1

Comments and Suggestions for Authors

This review is quite extensive and with a clear common thread. I believe that it will be very useful for readers who are starting in the field of plant-microorganism interaction and the state of the art of role of endophytes.

An interesting topic is the endophytic fungi genetic regulation inside and outside the host. The authors could also address the host transcriptional response when living with endophytic fungi.

Author answer: Thank you for your agreement and summary of our manuscript. I will also like to thanks for the suggestion to include the “host transcriptional response when living with endophytic fungi”. It is an interesting area but we think that the review paper is already very extensive with more than 250 references and if we include this section it will lose its focus on the topic. I think it could be a good topic for another review.

Reviewer 2 Report

Climate change and global trading trend in plant materials are fueling factors in the dissemination of plant pathogens. In order to protect plants and preserve the environment there is the need of bio-control management of plant pathogens instead of chemicals. Fungal endophytes are promising for this purpose. The authors did a concise review which will a long way help in the identification of research gaps and promising approach in addressing issues related to bio-control. Nevertheless, in my opinion I proposed some minor corrections which might help improve the write up. I urge the authors to address the comments as presented below.

General comments:

1- Legend for Figure 1 is mentioned on page 16 lines 632 to 648. I couldn't find this figure in the entire write up. Please kindly provide figure 1.

2- I think Table 1 should be improved. The column "Notable findings" can be deleted because the title and authors of the original findings are on this table. In addition, "see table in article", "see article" can be transfer to table legend represented by one or two asterisk, respectively. N/A should also be included in the table legend for it to standalone. 

Specific comments:

1- Line 20: ....review acts as......

2- Line 25: In agriculture,............

3- Lines 34 and 35 "Endophytes can cause damage to host under certain conditions" if this is true, then the sentence should be rephrased? 

4- Lines 103 to 106: I think this is a repetition of lines 25 to 27.

5- Lines 154: ...from selected taxonomic.........

6- Lines 159: ....may control plant pathogens......delete "for"

7- Lines 158 - 162: This describes methods in the detection of potential anti-pathogenic compounds and does not tie with the section's title. 

8- Lines 164: Colletotrichum acutatum and Sclerotium rolfsii are referred to as pathogenic species in this case, so should be in italics. 

9- Lines 177-179: In my opinion, this sentence does not change the sense or add extra information to the required content of this paragraph and should be left out.

10- Line 199: growth of...S. rolfsii (close the space).

11- Line 238: Correct to (..by 81%, 64%, and 70%.........)

12- Line 346: ..concentration of104....(space)

13- Lines 349-351: .....nematode motility deceased....!!! may be increased is the correct word.

14- Lines 359 and 362: Change the citation format.

15- Line 366: Close the space before the word "spp"

16- Line 367: Change nonalkaloid to non alkaloid.

17- Line 377: Close the space before the word "species".

18- Line 397: Close the space after the word "species".

19- Line 417: "Although is not within the scope of this review" This contradicts the objective stated in lines 15, 16, 97-101.

20- Line 422: Close the space after the word "with"

21- Line 652: Separate authors claim from the Funding section.

Author Response

Reviewer 2

Comments and Suggestions for Authors

Climate change and global trading trend in plant materials are fueling factors in the dissemination of plant pathogens. In order to protect plants and preserve the environment there is the need of bio-control management of plant pathogens instead of chemicals. Fungal endophytes are promising for this purpose. The authors did a concise review which will a long way help in the identification of research gaps and promising approach in addressing issues related to bio-control. Nevertheless, in my opinion I proposed some minor corrections which might help improve the write up. I urge the authors to address the comments as presented below.

Author answer: Thanks for your agreement and suggestions. We have answered your general comments as below.

General comments:

  • Legend for Figure 1 is mentioned on page 16 lines 632 to 648. I couldn't find this figure in the entire write up. Please kindly provide figure 1.

Author answer: We have added the figure to the manuscript but look like it did not uploaded properly due to the size of the file. We will make sure that Figure attach properly this time.

  • I think Table 1 should be improved. The column "Notable findings" can be deletedbecause the title and authors of the original findings are on this table. In addition, "see table in article", "see article" can be transfer to table legend represented by one or two asterisk, respectively. N/A should also be included in the table legend for it to standalone. 

Author answer: We have improved the table 1 and have reduced the text from the column of “Notable Finding”. We believe we should leave this column because otherwise it is simply a table of references. In its current format, it is more useful to users because it provides a brief summary of results that may spur interest. We have changed the table legend according to the reviewer’s suggestion and have added it to the table legend so it can stand alone.

Specific comments:

  • Line 20: ....review acts as......

Author answer: We have added “as” suggested by the reviewer.

  • Line 25: In agriculture,............

Author answer: We have added “comma” as suggested.

  • Lines 34 and 35 "Endophytes can cause damage to host under certain conditions"if this is true, then the sentence should be rephrased? 

Author answer: We have rephrased the sentence by adding most of the time at the end. Endophytes are microorganisms that live inside the plant for all or part of their life cycle while not causing damage or disease symptoms in their host most of the time.

  • Lines 103 to 106: I think this is a repetition of lines 25 to 27.

Author answer: We do not think it is the repetition, it is a good start for the section.

  • Lines 154: ...from selected........

Author answer: Thanks for pointing this out, it has been corrected.

  • Lines 159: ....may control plant pathogens......delete "for"

Author answer: We have deleted "for" as suggested.

  • Lines 158 - 162: This describes methods in the detection of potential anti-pathogenic compounds and does not tie with the section's title. 

Author answer: We believe that a very brief description of this workflow facilitates the reader’s understanding of how endophytes are subsequently screened and investigated for antifungal effects, which is applicable to the other sections further on. Therefore we opt to retain this text however we included it in the second paragraph of the section (Lines 122-144) to reduce repetition and improve the flow of ideas.

  • Lines 164: Colletotrichum acutatumand Sclerotium rolfsii are referred to as pathogenic species in this case, so should be in italics. 

Author answer : Thanks for paying attention to the small details. As suggested by the reviewer all scientific names has been italicized.

  • Lines 177-179: In my opinion, this sentence does not change the sense or add extra information to the required content of this paragraph and should be left out.

Author answer: We think this sentence is fine because it serves as a connection between the introductory sentence and listing examples of endophytic fusaria with anti-pathogen activities; it essentially states that a lot of work has been done with this genus but less so from an antifungal/agricultural perspective. We have rephrased the sentence as:

Many studies have investigated Fusarium metabolites for their application as pharmaceutical antimicrobial agents, but less focus has been placed on the antifungal properties of these compounds and their application in agricultural systems [80].

10- Line 199: growth of...S. rolfsii (close the space).

Author answer: We have removed the space.

11- Line 238: Correct to (..by 81%, 64%, and 70%.........)

Author answer: We have corrected as suggested.

12- Line 346: ..concentration of104....(space)

Author answer: We have corrected as suggested.

13- Lines 349-351: .....nematode motility deceased....!!! may be increased is the correct word.

Author answer: By “motility deceased” we means that nematode become less able to move and after 24 hour exposer the nematode died.

14- Lines 359 and 362: Change the citation format.

Author answer: We have fixed the issue.

15- Line 366: Close the space before the word "spp"

Author answer: We have removed the space.

16- Line 367: Change nonalkaloid to non alkaloid.

Author answer: We have corrected as suggested.

17- Line 377: Close the space before the word "species".

Author answer: We have removed the space.

18- Line 397: Close the space after the word "species".

Author answer: We have removed the space.

19- Line 417: "Although is not within the scope of this review" This contradicts the objective stated in lines 15, 16, 97-101.

Author answer: We have rephrased that part and now it is clear when we say this “Although it is not within the scope of this review, it has been shown that several fungal endophytes have anticancer and antiviral properties against human viruses [144–147]”.

20- Line 422: Close the space after the word "with"

Author answer: We have removed the space.

21- Line 652: Separate authors claim from the Funding section.

Author answer: We have separated the authors claim from the Funding section as suggested.

Reviewer 3 Report

The review paper by Grabka et al. reports on the beneficial role of endophytic fungi in agriculture, by means of their actions against fungal and bacterial pathogens, parasitic nematodes, viruses and mites. An extensive list of agriculture-friendly endophytes and their mode of action is provided, as well as several examples of successful application in agriculture.

The topic of this review is of outstanding interest for a modern and sustainable agriculture, since it highlights the concrete possibility to substitute dangerous agrochemicals with environmental friendly tools in biological pests control, such as the use of either specific endophytic fungi, endophyte consortia or whole microbiomes.

To my opinion this review is clearly written, comprehensive and of relevance to the field, although similar reviews were published recently. The statements and conclusions drawn are coherent and supported by the listed citations.

I have only some minor comments and changes to suggest:

L32: delete “an”

L153: pharmaceuticals

L199: delete the space between ‘of’ and ‘S.’ Check for spacing also L360,366,397,422 and elsewhere

L201-202: please check character dimensions here and throughout the manuscript

L224: did you mean Trichoderma harzianum? Please correct here and below

L315-319: This list of genera could be completed with Talaromyces, Arthrobotrys, Monacrosporium, etc. as suggested in: Aït Hamza, M.; Lakhtar, H.; Tazi, H.; Moukhli, A.; Fossati-Gaschignard, O.; Miché, L.; Roussos, S.; Ferji, Z.; El Mousadik, A.; Mateille, T.; et al. Diversity of Nematophagous Fungi in Moroccan Olive Nurseries: Highlighting Prey-Predator Interactions and
Efficient Strains against Root-Knot Nematodes. Biol. Control 2017, 114, 14–23.

L633: Figure 1 is not present in the manuscriopt, please provide it.

Author Response

Reviewer 3

Comments and Suggestions for Authors

The review paper by Grabka et al. reports on the beneficial role of endophytic fungi in agriculture, by means of their actions against fungal and bacterial pathogens, parasitic nematodes, viruses and mites. An extensive list of agriculture-friendly endophytes and their mode of action is provided, as well as several examples of successful application in agriculture.

The topic of this review is of outstanding interest for a modern and sustainable agriculture, since it highlights the concrete possibility to substitute dangerous agrochemicals with environmental friendly tools in biological pests control, such as the use of either specific endophytic fungi, endophyte consortia or whole microbiomes.

To my opinion this review is clearly written, comprehensive and of relevance to the field, although similar reviews were published recently. The statements and conclusions drawn are coherent and supported by the listed citations.

Author answer: We thanks the reviewer for the encouragement and liking our review article. Please see the answers to the minor comments and suggestion.

I have only some minor comments and changes to suggest:

L32: delete “an”

Author answer: We have deleted “an” as suggested.

L153: pharmaceuticals

Author answer: We have deleted “pharmaceuticals” as suggested.

L199: delete the space between ‘of’ and ‘S.’ Check for spacing also L360,366,397,422 and elsewhere

Author answer: The space has been removed from all places.

L201-202: please check character dimensions here and throughout the manuscript

Author answer: We have changed the character dimensions.

L224: did you mean Trichoderma harzianum? Please correct here and below

Author answer: Thanks for the spelling correction. We have corrected it throughout the manuscript.

L315-319: This list of genera could be completed with Talaromyces, Arthrobotrys, Monacrosporium, etc. as suggested in: Aït Hamza, M.; Lakhtar, H.; Tazi, H.; Moukhli, A.; Fossati-Gaschignard, O.; Miché, L.; Roussos, S.; Ferji, Z.; El Mousadik, A.; Mateille, T.; et al. Diversity of Nematophagous Fungi in Moroccan Olive Nurseries: Highlighting Prey-Predator Interactions and
Efficient Strains against Root-Knot Nematodes. Biol. Control 2017, 114, 14–23.

Author answer: We have added these genera to the list as suggested by the reviewer and have added the reference recommended: Hamza, M.A., Lakhtar, H., Tazi, H., Moukhli, A., Fossati-Gaschignard, O., Miché, L., Roussos, S., Ferji, Z., El Mousadik, A., Mateille, T. and Boubaker, H., 2017. Diversity of nematophagous fungi in Moroccan olive nurseries: Highlighting prey-predator interactions and efficient strains against root-knot nematodes. Biological Control114, pp.14-23.

L633: Figure 1 is not present in the manuscript, please provide it.

Answer: We have added the figure to the manuscript but look like it did not uploaded properly due to the size of the file. We will make sure that Figure 1 attach properly this time.

Reviewer 4 Report

see remarks in the file.

some specific remarks:

Way wasn’t this review submitted to journal of fungi belonging to MDPI? I think it is more suited to it than plants.

Line 78. As much as I know this statement is correct only for herbaceous plants as far as for trees there are no examples of endophytes that spread systemically. Please give a reference that contradict me if you think differently. As for the paper referenced by Chotulu et al 2018 it refer a paper by Verma et al 2011 were it states that the Fusarium isolated form the roots was isolated from the leaves as well yet they give no prof for its movement in the trees tissue from the root to the leaves.

Line 102-259. the title of this paragraph is a bit misleading. The content of this paragraph does not give description of the role of fungal endophytes against fungal pathogen. It gives a list of fungal endophytes found to have anti-fungal properties mainly due to secretion of secondary metabolites with bioactive abilities. I believe the title should be modified to a more descriptive form of the paragraphs content.

Please consider rephrasing to something like:” Biologically active fungal endophytes and their effects on fungal pathogens”

Line 260-294. same as for the remark for lines 102-259

Line 295-414. same for the title.

Line 415. the title and the first sentences give the reader the impression that you are going to discuss a direct effect or roll of endophytes against viruses. This is not the case; the paragraph is mentioning studies were endophytes n were influencing the viruses vectors (insects) or a systemic plant defense activation that resulted in reduced virus infection. This is fine but it needs to be clear at the beginning of the paragraph that the effects against viruses are indirect.

Line 423. this is not an antiviral effect but an anti insect effect with un direct effect on the presence of viruses  

Line 633. Legend to the figures

I could not find a figure attached to the manuscript. There is no in text reference to a figure. And the legend does not correlate to any of the text in the manuscript by its content.

To my opinion, the manuscript does not renew or add more or beyond many other reviews dedicated to endophytes. Most of the content in its chapters are quite superficial. The titles of the chapters are in some manner misleading as the titles of chapters 2-6 clam to deal with “the role of endophytes” against the various pathogens but in practice the chapters do not deal with the role of these endophytes. Instead, the readers find description and examples of endophytes exhibiting activity against the specific pathogens (perhaps except in the case of nematodes). Therefore, I think it is necessary to change the titles of these chapters so that it will be more descriptive of the content of the chapter. The attached table can be helpful for anyone looking for information available in the literature on the subject or regarding a particular type of fungal endophyte. At the end of the manuscript, there is a title for a figure that I could not find and I do not quite understand the relevance of what is described there to the content of the article. There is also no reference to the figure in the text itself.

Author Response

Reviewer 4

Comments and Suggestions for Authors

see remarks in the file.

some specific remarks:

Way wasn’t this review submitted to journal of fungi belonging to MDPI? I think it is more suited to it than plants.

Author answer: Thanks for the suggestion to submit the paper to the journal of fungi but we think it is more related to this special issue of “Plant Interaction with Fungal Endophytes”

Line 78. As much as I know this statement is correct only for herbaceous plants as far as for trees there are no examples of endophytes that spread systemically. Please give a reference that contradict me if you think differently. As for the paper referenced by Chotulu et al 2018 it refer a paper by Verma et al 2011 were it states that the Fusarium isolated form the roots was isolated from the leaves as well yet they give no prof for its movement in the trees tissue from the root to the leaves.

 Author answer:  Thanks for raising this point. We have modified the statement and added some new references to support the argument.

Line 102-259. the title of this paragraph is a bit misleading. The content of this paragraph does not give description of the role of fungal endophytes against fungal pathogen. It gives a list of fungal endophytes found to have anti-fungal properties mainly due to secretion of secondary metabolites with bioactive abilities. I believe the title should be modified to a more descriptive form of the paragraphs content.

Please consider rephrasing to something like:” Biologically active fungal endophytes and their effects on fungal pathogens”

Author answer:As suggested by the reviewer we have rephrased the heading accordingly.

Line 260-294. same as for the remark for lines 102-259

Author answer: As suggested by the reviewer we have rephrased the heading accordingly.

Line 295-414. same for the title.

Author answer:  As suggested by the reviewer we have rephrased the heading accordingly.

Line 415. the title and the first sentences give the reader the impression that you are going to discuss a direct effect or roll of endophytes against viruses. This is not the case; the paragraph is mentioning studies were endophytes n were influencing the viruses vectors (insects) or a systemic plant defense activation that resulted in reduced virus infection. This is fine but it needs to be clear at the beginning of the paragraph that the effects against viruses are indirect.

Author answer:  As suggested by the reviewer we have rephrased the heading accordingly. We have also stated that most of the antiviral protection by endophytes is indirect either against the viral vector or by increasing the plant defence against the viruses.

Line 423. this is not an antiviral effect but an anti insect effect with un direct effect on the presence of viruses.  

Author answer: We have added a sentence to make it clear that it is an indirect effect of endophyte on viral vectors and not a direct antiviral effect.

Line 633. Legend to the figures

I could not find a figure attached to the manuscript. There is no in text reference to a figure. And the legend does not correlate to any of the text in the manuscript by its content.

Author answer: We have added the figure to the manuscript but it maybe not loaded properly due to the size of the file. We will make sure that it attach properly this time. We have added the reference to figure in the text.

 To my opinion, the manuscript does not renew or add more or beyond many other reviews dedicated to endophytes. Most of the content in its chapters are quite superficial. The titles of the chapters are in some manner misleading as the titles of chapters 2-6 clam to deal with “the role of endophytes” against the various pathogens but in practice the chapters do not deal with the role of these endophytes. Instead, the readers find description and examples of endophytes exhibiting activity against the specific pathogens (perhaps except in the case of nematodes). Therefore, I think it is necessary to change the titles of these chapters so that it will be more descriptive of the content of the chapter. The attached table can be helpful for anyone looking for information available in the literature on the subject or regarding a particular type of fungal endophyte. At the end of the manuscript, there is a title for a figure that I could not find and I do not quite understand the relevance of what is described there to the content of the article. There is also no reference to the figure in the text itself.

Author answer: We have changed the tittle of the manuscript and also the tittle of the sub-heading to better describe the content of the chapter.

Reviewer 5 Report

Comments to the manuscript plants-1530331 "Fungal endophytes and their application in agriculture for pest and disease management".

Authors propose a complete and up to date review of the scientific literature of the befefits of the fungal endophytes on plant metabolisms. The application possibilities in agriculture for pest and disease management are now only partially viable and mostly potential. However, the work is well organized, with a large list of citations and a useful Table of synthetic references. Unfortunately the Figure 1 was not available in the submitted manuscript.

In my opinion, the manuscript is suitable for publication after some minor changes. Please check the text between lines 163 and 175 alla the scientific names should be reported in italics.

Author Response

Reviewer 5

Comments and Suggestions for Authors

Comments to the manuscript plants-1530331 "Fungal endophytes and their application in agriculture for pest and disease management".

Authors propose a complete and up to date review of the scientific literature of the befefits of the fungal endophytes on plant metabolisms. The application possibilities in agriculture for pest and disease management are now only partially viable and mostly potential. However, the work is well organized, with a large list of citations and a useful Table of synthetic references. Unfortunately the Figure 1 was not available in the submitted manuscript.

In my opinion, the manuscript is suitable for publication after some minor changes. Please check the text between lines 163 and 175 alla the scientific names should be reported in italics.

Author answer: Thanks for your agreement and suggestions. We also thanks you for paying attention to the small details. As suggested by the reviewer all scientific names has been italicized.

We have added the figure to the manuscript but look like it did not uploaded properly due to the size of the file. We will make sure that Figure attach properly this time.

Round 2

Reviewer 4 Report

no comments